# *Guiclupea superstes*, gen. et sp. nov., the youngest ellimmichthyiform (clupeomorph) fish to date from the Oligocene of South China

Gengjiao Chen[1], Mee-mann Chang[2,3], Feixiang Wu[2,4] and Xiaowen Liao[1]

[1] Natural History Museum of Guangxi Zhuang Autonomous Region, Nanning, China
[2] Key Laboratory of Vertebrates, Evolution and Human Origins of Chinese Academy of Sciences IVPP, CAS, Beijing, China
[3] College of Earth and Planetary Sciences, University of Chinese Academy of Sciences, Beijing, China
[4] CAS Center for Excellence in Life and Paleoenvironment, Beijing, China

Corresponding author
Gengjiao Chen,
cgengjiao@aliyun.com

## ABSTRACT

A new ellimmichthyiform, *Guiclupea superstes*, gen. et sp. nov., from the Oligocene Ningming Formation of Ningming Basin, Guangxi Zhuang Autonomous Region, South China is described herein. With relatively large body size, parietals meeting at the midline, anterior ceratohyal with a beryciform foramen in the center, a complete predorsal scutes series of very high number and about equally-size scutes with radiating ridges on dorsal surface, first preural centrum unfused with first uroneural but fused with the parhypural, and first ural centrum of roughly the same size as the preural centrum, *Guiclupea superstes* cannot be assigned to the order Clupeiformes. The phylogenetic analyses using parsimony and Bayesian inference methods with *Chanos*/*Elops* as outgroup respectively suggests that the new form is closer to ellimmichthyiform genus *Diplomystus* than to any other fishes, although there are some discrepancies between the two criteria and different outgroups used. It shares with *Diplomystus* the high supraoccipital crest, pelvic-fin insertion in advance of dorsal fin origin, and the number of predorsal scutes more than 20. The new form represents the youngest ellimmichthyiform fish record in the world. Its discovery indicates that the members of the Ellimmichthyiformes had a wider distribution range and a longer evolutional history than previously known.

## INTRODUCTION

The Ellimmichthyiformes is one of the two major clades of the Clupeomorpha (*Nelson, Grande & Wilson, 2016*). The Recent Clupeomorpha is represented only by the order Clupeiformes, which is amongst the most economically important fish species for food, and contains both fossil and extant herrings, anchovies, and other relatives (*Lavoué, Konstantinidis & Chen, 2014*). The order Ellimmichthyiformes is an extinct cosmopolitan clade (*Nelson, Grande & Wilson, 2016*), established by *Grande* in *1982*. It initially included only a single family Paraclupeidae (see *Chang & Grande, 1997*; *Hay et al., 2007*) with only

two genera–*Diplomystus* and *Ellimmichthys*, diagnosed by bearing a series of predorsal scutes expanding laterally then taking a subrectangular-shape, and lacking some derived characters of the Clupeiformes, e.g., presence of recessus lateralis, parietal bones completely separated by the supraoccipital, and loss of the 'beryciform' foramen in anterior ceratohyal (*Grande, 1982*; *Grande, 1985*). Since the establishment of Ellimmichthyiformes, especially in the last two decades, many new and previously known genera and species have been either added or moved to this order (*Silva Santos, 1994*; *Silva Santos, 1990*; *Silva Santos, 1994*; *Bannikov & Bacchia, 2000*; *Chang & Maisey, 2003*; *Poyato-Ariza, López-Horgue & García-Garmilla, 2000*; *Forey et al., 2003*; *Hay et al., 2007*; *Alvarado-Ortega, Ovalles-Damián & Arratia 2008*; *Khalloufi, Zaragüeta-Bagils & Lelièvre, 2010*; *Newbrey et al., 2010*; *Murray & Wilson, 2011*; *Malabarba et al., 2004*; *Bannikov, 2015*; *Vernygora & Murray, 2015*; *Murray et al., 2016*; *Alvarado-Ortega & Melgarejo-Damián, 2017*; *Polck et al., 2020*; etc.), although some of them have the predorsal scutes pattern only partially in agreement with or completely disagree with this order-level character, e.g., *Ellimma branneri*, whose anterior predorsal scutes are longer than broad; *Scutatospinosus itapagipensis* and *Codoichthys carnavalii*, without any subrectangular predorsal scutes at all, completely disagree with the order-level character. Along with the increasing membership of this order, several families were erected, and the interest in the definition, classification, and intra-relationship of the group has been increasing (*Bannikov & Bacchia, 2000*; *Chang & Maisey, 2003*; *Zaragüeta-Bagils, 2004*; *Alvarado-Ortega, Ovalles-Damián & Arratia, 2008*; *Murray & Wilson, 2013*; *Figueiredo & Ribeiro, 2016*; *Vernygora, Murray & Wilson, 2016*; *Marramà & Carnevale, 2017*; *Boukhalfa et al., 2019*; *Vernygora & Murray, 2020*; etc.), although no definitive consensus has been reached on these issues. The main differences among the results of previous studies are the relationship of *Armigatus* and *Diplomystus*. Some analyses suggested that *Armigatus* is sister to *Diplomystus* (*Chang & Maisey, 2003*; *Murray & Wilson, 2013*) whereas others suggested that *Armigatus* is in a more basal (*Forey, 2004*; *Figueiredo & Ribeiro, 2016*) or derived (*Vernygora, Murray & Wilson, 2016*; *Marramà & Carnevale, 2017*; *Boukhalfa et al., 2019*) position than *Diplomystus*, or *Armigatus* is not an ellimmichthyiform member (*Zaragüeta-Bagils, 2004*). And still others suggested that *Ornategulum* may or may not be an ellimmichthyiform (*Murray & Wilson, 2011*; *Marramà et al., 2019*; *Figueiredo & Ribeiro, 2017*; *Boukhalfa et al., 2019*). To date, the reported members of the group have reached 21 genera and 38 species at least, ranging from the Early Cretaceous to middle Eocene marine and non-marine strata of Eastern Asia, Middle East, North and South America, Africa, and Europe (Fig. 1). No ellimmichthyiform fish from strata younger than the Eocene has ever been reported. Accordingly, it was believed that the ellimmichthyiforms finally became extinct after the middle Eocene. However, recently, an Oligocene ellimmichthyiform fish was discovered from the non-marine deposits of Ningming Basin, Guangxi Zhuang Autonomous Region, South China. Therefore, this new ellimmichthyiform fish represents the youngest record. This discovery not only extends the spatial and temporal distribution of ellimmichthyiforms, but also sheds new light on our understanding of the evolutionary history and paleobiogeography of the order. Herein we describe the new form, perform a phylogenetic analysis of the Ellimmichthyiformes, and discuss its taxonomic position and paleobiographical implications.

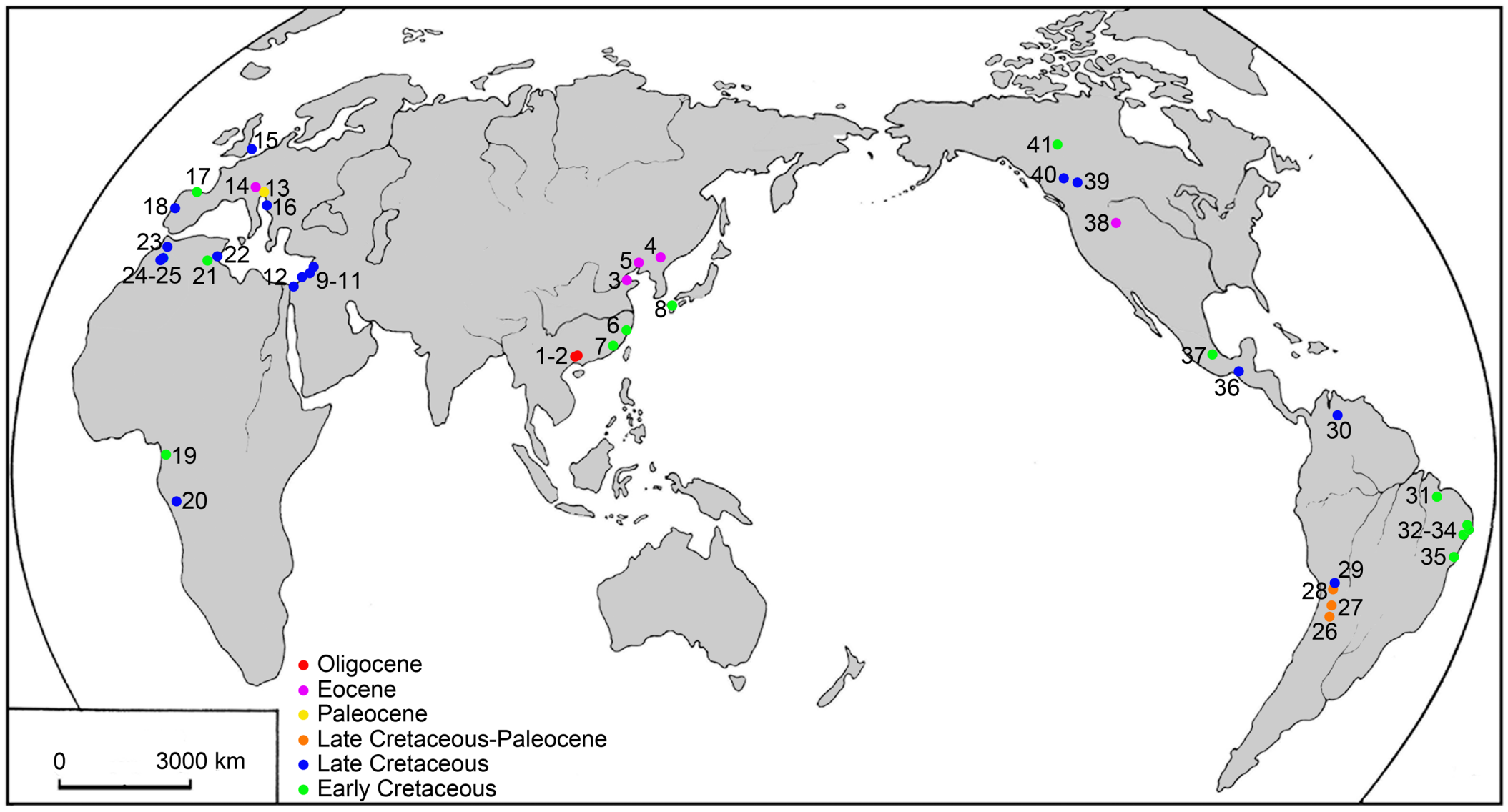

**Figure 1 Map of the main localities of ellimmichthyiform fossils.** 1. Ningming, 2. Nanning, Guangxi, 3. Kenli, Shandong, 4. Huadian, Jilin, 5. Xialiaohe Oilfield, Liaoning, 6. Linhai, Zhejiang, and 7. Anxi, Fujian, China; 8. northern Kyushu, Japan; 9. Namoura, 10. Hakel and Hajula, Mount Lebanon, and 11. Sahel Alma, Lebanon; 12. Ein Yabrud, Palestine; 13. Trieste and 14. Bolca Lagerstätte, Italy; 15. Kent, England; 16. Dalmatia, Croatia; 17. Basque-Cantabrian Basin, Spain; 18. Portugal; 19. Equatorial Guinea; 20. Kwango, Zaire; 21. Chotts Basin and 22. Gabès, Tunisia; 23. Jbel Tselfat, 24. Aoult, and 25. Jbel Oum Tkout, Morocco; 26. Sierra de Santa Bárbara and 27. La Puerta, Argentina; 28. Cayara and 29. Agua Clara, Bolivia; 30. Santa Barbara, Venezuela; 31. São Luís-Grajaú Basin, 32. Alagoas Basin, 33. Sergipe Basin, 34. Recôncavo Basin, and 35. Santos Basin, Brazil; 36. Chiapas and 37. Puebla, Mexico; 38. Wyoming, USA; 39, near Red Deer River, 40. south of Grand Prairie, and 41. Northwest Territories, Canada.

## MATERIAL AND METHODS

The fossil specimens except NHMG 038777, that are described herein, include both articulated skeletons and detached bones and were collected from the outcrops about 2.5∼3.5 km west of Ningming County, Guangxi, South China (Fig. 1), about 40 km away from the boundary of China and Vietnam and about 120 km northwest of the South China Sea. They are housed in the NHMG now. The fossil-bearing strata is positioned in the middle-upper part of the Second Member of the Yongning Group (*Bureau of Geology and Mineral Resources of Guangxi Zhuang Autonomous Region, 1985*), or Ningming Formation (*Li, Qiu & Li, 1995*), which is a set of fossiliferous lacustrine sediments dominated by light-gray, yellowish mudstones, occasionally containing fine sand grains. The deposits also contain a variety of cyprinid and a few siluriform and perciform fishes, and a large number of plant macrofossils (*Chen, Liu & Chang, 2018*). No isotopically datable volcanic material was found at the locality. The geological age of the Ningming Formation, according to palynologists (*Wang et al., 2003*), is Oligocene. Paleobotanists concurred after they had studied macrofossil plant from the same strata (*Li et al., 2003*; *Shi, Zhou & Xie, 2010*; *Shi, Zhou & Xie, 2012*; *Shi, Xie & Li, 2014*; *Wang et al., 2014*; *Dong et al., 2017*; *Ma et al., 2017*; etc.). We applied this geological age also when we studied *Huashancyprinus robustispinus*

(Cyprinidae, Cypriniformes) from the same locality and horizon (*Chen & Chang, 2011*) and adopt it herein. NHMG 038777 is a disarticulated dentary collected from the Yongning Formation of Santang, Nanning basin, Guangxi. The geological age of Yongning Formation is the Oligocene (*Zhao, 1983*; *Zhao, 1993*; *Quan et al., 2016*).

Fossil fish materials were prepared mechanically with steel needles of different sizes under a binocular microscope. Line drawings were done based on the observations of the fossils under an Olympus SZ61 microscope and the photos.

The taxonomic terminology and the methods of counting and measurement used here follow *Grande (1982)* and *Forey et al. (2003)*. The descriptions of gill rakers follow *Bornbusch & Lee (1992)*. Specimens used for comparison include: (1) *Paraclupea chetungensis* (*Sun, 1956*), including IVPP V816, V2986.2, V3002.1, 5-8, 10, 12, 15, 19, from the Lower Cretaceous Chawan Formation, eastern China; (2) *Diplomystus shengliensis* (*Zhang, Zhou & Qing, 1985*), including SOF 790001, SOF 790002, and SOF 790003, and (3) *Knightia bohaiensis* (*Zhang, Zhou & Qing, 1985*), including SOF 790003, from the top of series 4 to the bottom of series 3 of the Shahejie Formation, Middle Eocene, East China; and 4) dried skeleton and disarticulated bones of *Ilisha elongate*, NHMG 038785, collected from Nanning Dancun Market.

The electronic version of this article in portable document format will represent a published work according to the International Commission on Zoological Nomenclature (ICZN), and hence the new names contained in the electronic version are effectively published under that Code from the electronic edition alone. This published work and the nomenclatural acts it contains have been registered in ZooBank, the online registration system for the ICZN. The ZooBank Life Science Identifiers (LSIDs) can be resolved and the associated information viewed through any standard web browser by appending the LSID to the prefix http://zoobank.org/. The LSID for this publication is: urn:lsid:zoobank.org:pub:99B7F0EE-3695-4178-9606-1CD8BD90316C. The online version of this work is archived and available from the following digital repositories: PeerJ, PubMed Central, and CLOCKSS.

The phylogenetic analyses are based on a data matrix (see Appendices S1–S3) consisting of 55 morphological characters and 40 taxa, including three Recent clupeiform species (*Denticeps clupeoides*, *Chirocentrus dorab*, and *Odaxothrissa vittata* (the first one is the only extant member of the Denticipitoidei; the latter two represent the Clupeoidei), a gonorynchiform or elopomorph (*Chanos chanos* or *Elops saurus*, being used as outgroup alternatively to polarize the characters and root the tree), the enigmatic fossil *Ornategulum sardinioides* Forey 1973, and our new form (to test its position within the Clupeomorpha). Characters are adopted mainly from *Chang & Maisey (2003)* and *Murray & Wilson (2013)*.

The analyses use both parsimony and Bayesian inference methods, for both methods have advantages and disadvantages for morphological data (*Bai et al., 2020*). The parsimony analyses were performed with TNT 1.5 (*Goloboff, Farris & Nixon, 2008*), using the Traditional Search method with 1000 replicates and tree bisection and reconnection (TBR) swapping algorithm. All characters are unordered and equally weighted. The most parsimonious trees (MPTs) generated by the analysis were used to construct a strict

consensus tree. Tree length, consistency index (CI), retention index (RI), Bremer support and bootstrap values were then calculated for the strict consensus tree.

Bayesian analyses were conducted by MrBayes 3.1.2 (*Ronquist & Huelsenbeck, 2003*). For the substitution models, the Mkv model was used with an assumption of gamma rate variation across characters. Markov chain Monte Carlo analysis consists of four chains, which were run simultaneously with 2,000,000 trees, sampling 1/100 trees, with a burn-in value of 5,000. The remaining trees were used to build a 50% majority rule consensus tree, and statistical support of each node was assessed by posterior probabilities.

## Systematic Paleontology

Infraclass TELEOSTEI *Muller, 1845*
Cohort CLUPEOCEPHALA *Patterson & Rosen, 1977*
Superorder CLUPEOMORPHA *Greenwood et al., 1966*
Order ELLIMMICHTHYIFORMES *Grande, 1982*

*Diplomystus* clade
Genus *GUICLUPEA* gen. nov.

**Diagnosis**: A fairly large-sized, double-armored ellimmichthyiform fish, differing from other genera of the order in the following combination of characters: dorsal body margin without marked angle at the dorsal fin insertion; posttemporal large; predorsal scutes series complete, with scutes small, numerous (about 55), all about equal in size, and with ridges on dorsal surface; number of predorsal bones ten or more; no diastema between second and third hypural; proximal end of middle principal caudal fin rays enlarged.

**Etymology**: 'gui', spelling or pingyin of the Chinese character '桂', the abbreviation in Chinese of the Guangxi Zhuang Autonomous Region, a province of China from where the fossil materials were collected; 'clupea', from the Latin, to indicate clupeomorph affinities of the new taxon.

**Type species**: *Guiclupea superstes* gen. et sp. nov., the only known species.

*Guiclupea superstes* gen. et sp. nov.
(Figs. 2–7)

**Diagnosis**: See generic diagnosis. Pectoral fin rays 18, pelvic fin rays 5~6, dorsal fin rays 14, anal fin rays 38, total number of vertebrae about 46, 23 caudal and two ural vertebrae.

**Etymology**: 'superstes', Latin 'survivore'. The species name means that the species survived in the Oligocene when all members of the order Ellimmchthyiformes had become extinct.

**Holotype**: NHMG 005532, a nearly complete skeleton, part and counterpart (Figs. 2A–2B).

**Paratypes**: NHMG 033659 (Fig. 3A), a relatively complete skeleton with the snout and the caudal fin rays missing; NHMG 033658, a skeleton from the anterior margin of the

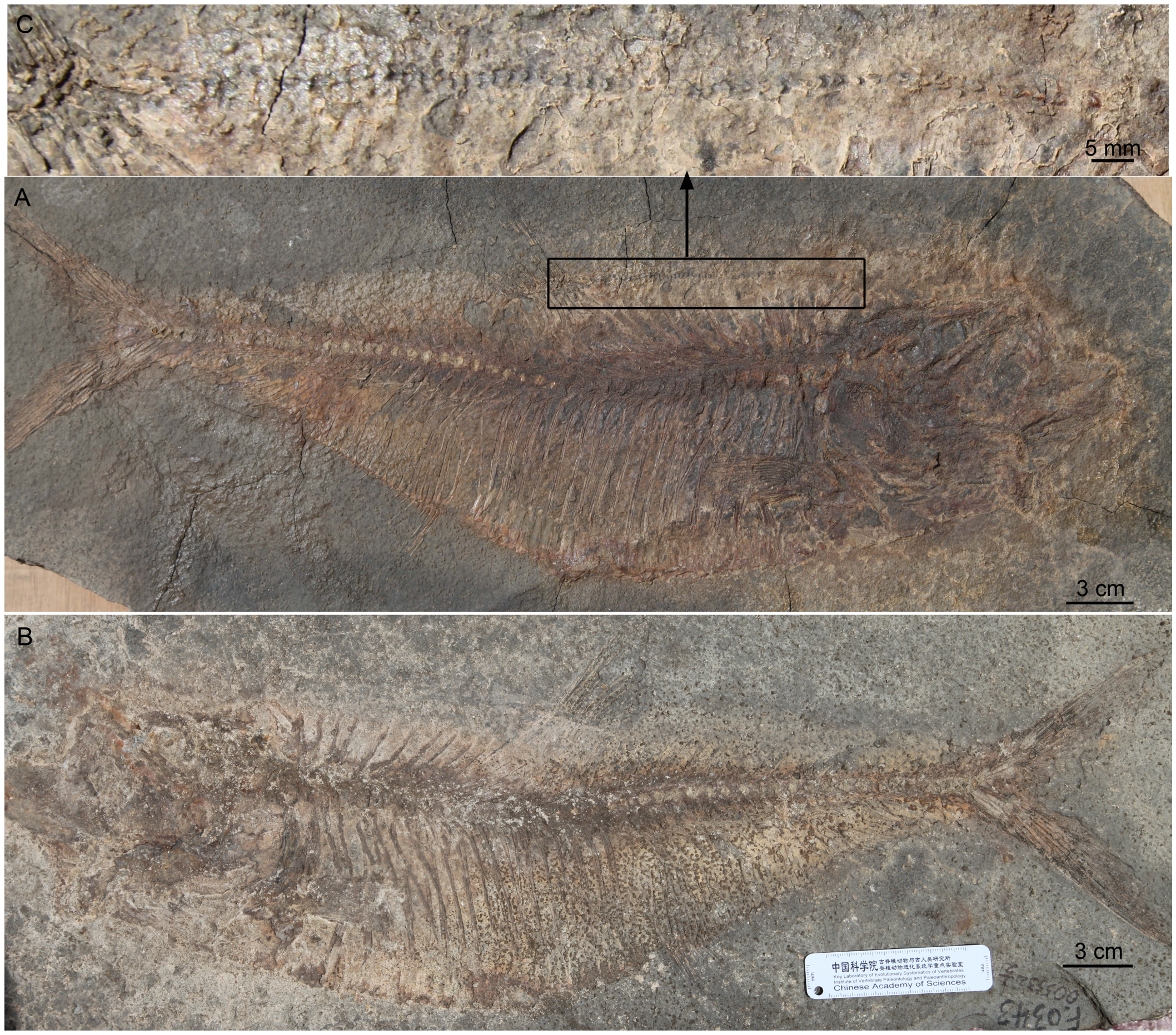

**Figure 2** ***Guiclupea superstes*, gen. et sp. nov.** (A) and (B) photograph of the holotype (NHMG 005532); (C) close up of the complete predorsal scutes series maked by the black box in (A).

orbital to the caudal peduncle, part of the anterior portion (Fig. 6) and counterpart of the posterior portion; NHMG 011648 (Fig. 7A), caudal peduncle to caudal fin, part and counterpart.

**Additional material**: NHMG 033660 (Fig. 5), disarticulated bones of the skull and anterior part of the body; NHMG 004929, an incomplete skeleton with the posterior part of the body missing; NHMG 033661, disarticulated bones of the skull, the anterior ceratohyal and entopterygoid in this specimen are shown in Fig. 4F; NHMG 033680, disarticulated bones of the skull and anterior part of the body, the posterior ceratohyals in this specimen are shown in Fig. 4G; NHMG 033685, premaxilla (Fig. 4C); NHMG 038778, an incomplete skeleton with the head and caudal skeleton and fin missing; NHMG 033681–033683 and NHMG 038777, dentary (NHMG 033682 and 033683 shown in Figs. 4D and 4E respectively); NHMG 011647 and NHMG 011649, caudal skeleton and caudal fin; NHMG 011650–011651, caudal skeleton.

**Localities and horizon**: Gaoling Village (22°07N, 107°02′E), Ningming County, and Santang (22°52′N, 108°25′E), Nanning, Guangxi province, China; middle to upper portion

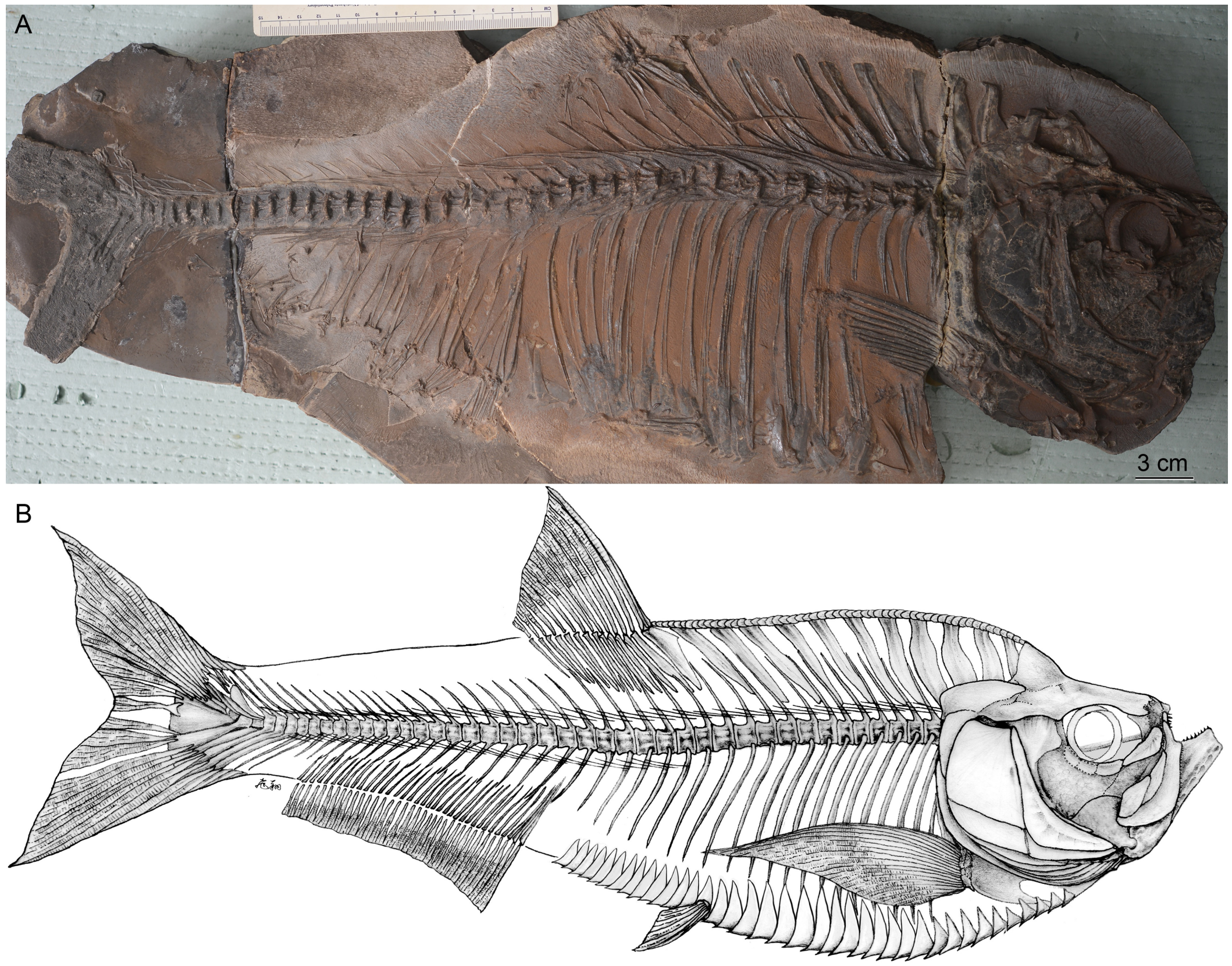

**Figure 3** ***Guiclupea superstes*, gen. et sp. nov.** (A) photograph of the NHMG 033659; (B) tentative restoration mainly based on the holotype and paratypes.

of the second member of the Yongning Group or Ningming Formation and Yongning Formation; Oligocene.

## Description

(1) **General Appearance**. This new form is a fairly large-sized double-armored clupeomorph. The total length of the holotype (Figs. 2A–2B) is about 526 mm. In the known largest incomplete specimen, NHMG 033659 (Fig. 3A), the preserved portion reaches 638 mm in length, and the distance from the anterior margin of the lacrimal to the caudal fin base is about 585 mm. The fish has an elongate fusiform body. The standard length in the holotype is about 3.3 times the maximum body depth. The anterior dorsal margin of the body is rounded and convex, without a marked angle at the origin of the dorsal fin as is typical in paraclupeids. The ventral border in front of the insertion of the pelvic fin is also convex, but is straight and rising obliquely upwards behind the insertion, making the posterior part of the body gradually narrower caudally. The origin of the dorsal fin is posterior to the level of the insertion of the pelvic. The anal fin has a long base. The caudal fin is deeply forked. There is a complete series of predorsal and ventral scutes along the dorsal and ventral margins of the body. The meristic characters are listed in Table 1.

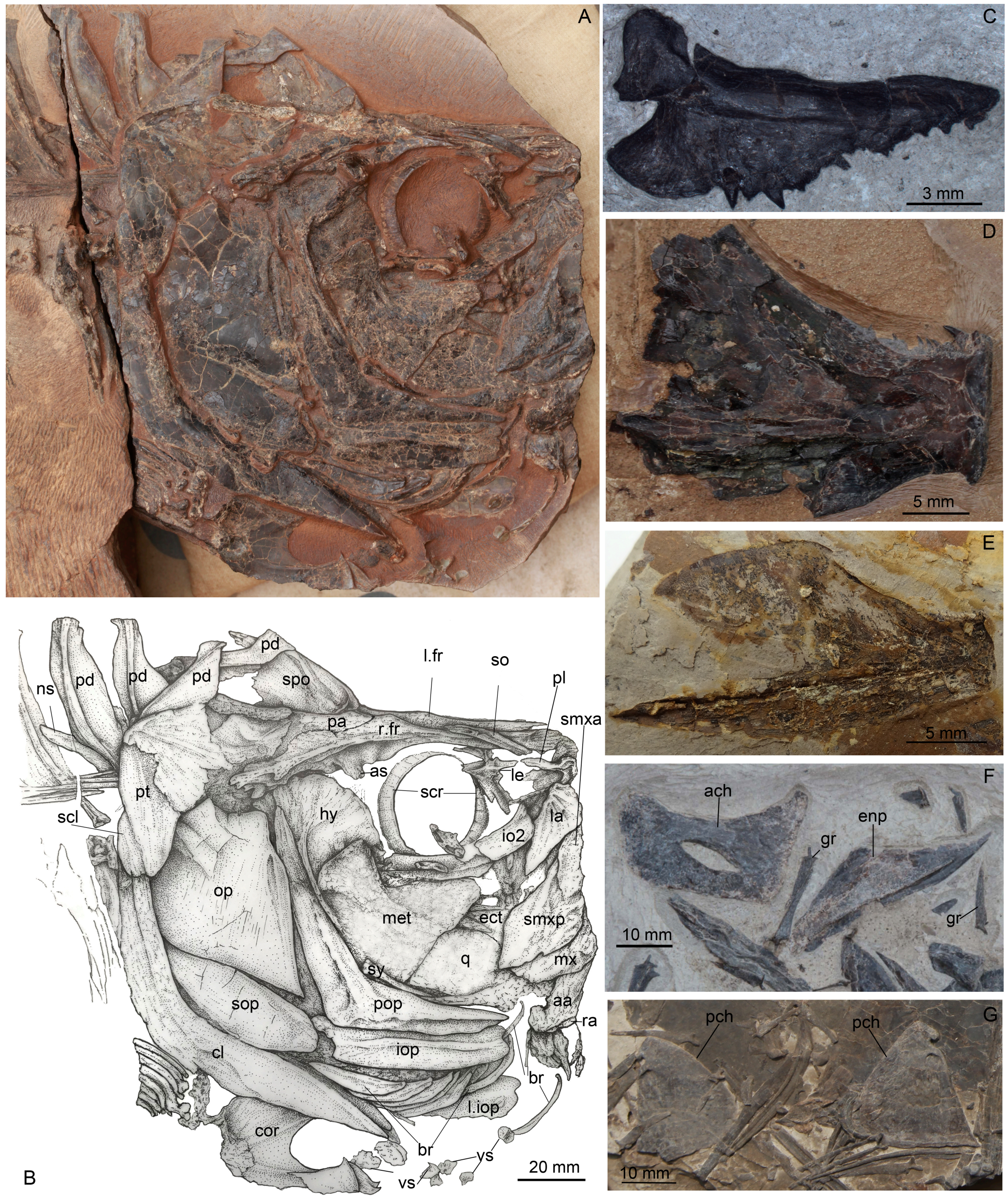

**Figure 4** ***Guiclupea superstes*, gen. et sp. nov.** (A) photograph, and (B) line drawing of the head in NHMG 033659. Anterior facing right. (C) photograph of a left premaxilla, NHMG 033685; (D) photograph of an incomplete dentary, NHMG 033682, showing the oral teeth; (E) photograph of a dentary, NHMG 033683; (F) photograph, showing anterior ceratohyal and entopterygoid in NHMG 033661; (G) photograph, showing posterior ceratohyals in NHMG 033680.

(2) **Skull Roof.** The head is slightly longer than deep (Figs. 2A; 4A). The skull roof above the orbit is narrow. The frontal is a long bone, with its posterior one-fourth expanding laterally; a longitudinal ridge for the supraorbital sensory canal runs along its dorsal surface (Figs. 4A–4B; 5). The frontal sutures with the anterior edge of the parietal posteriorly.

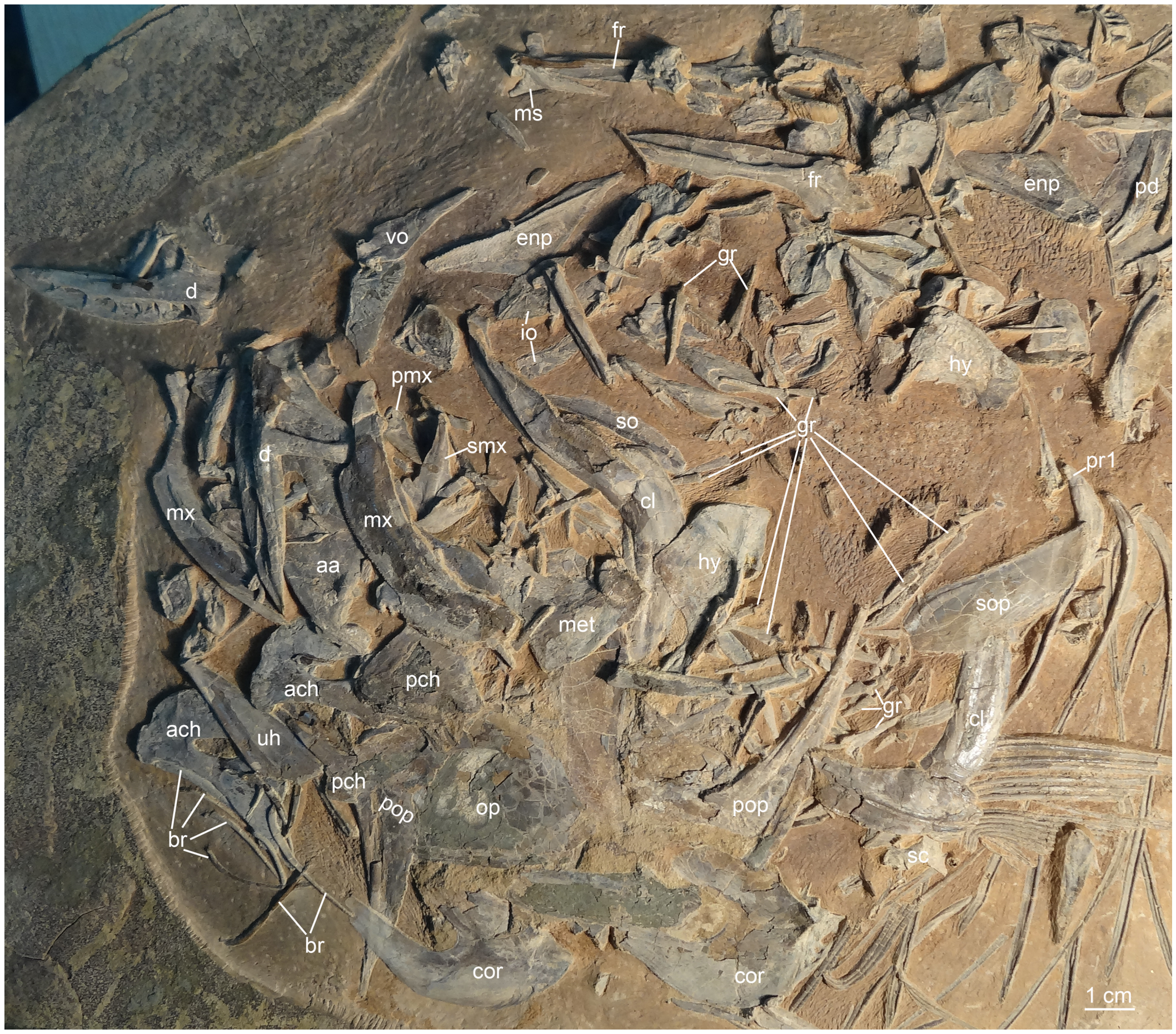

**Figure 5** ***Guiclupea superstes*, gen. et sp. nov.** Photograph of the disarticular skull and anterior trunk bones of NHMG 033660.

Two parietal bones meet at the midline, at least in the anterior part (Figs. 4A–4B), as in primitive clupeomorphs (*Grande, 1985*). No fontanelle between the anterior part of the frontals is observed, which is often present in clupeoids and *Paraclupea chetungensis* (*Chang & Grande, 1997*). Anterior to the frontal is the mesethmoid, which bears a lateral process on each side (Fig. 5). The lateral ethmoid, contacting the frontal at its anteriolateral margin, is situated anterior to the orbit, forming the lateral portion of the anterior wall of the orbit (Figs. 4A–4B). The outlines of the pterotic and sphenotic are not clear, but the strong ventrally directed process of the autosphenotic can be seen, lying in front of the head of the hyomandibula (Figs. 4A–4B). The supraoccipital is situated posteriorly and sutures with the parietals anteriorly. The supraoccipital crest is well-developed, high and triangular, making the lateral profile of the skull roof a distinct angle between the anterior and the posterior parts (Figs. 4A–4B). The external surface of all the skull roof bones lacks ornamentation, except for a longitudinal ridge containing the supraorbital sensory canal (Figs. 2A–2B, 4A, 5, 6A). No openings of the recessus lateralis are observed.

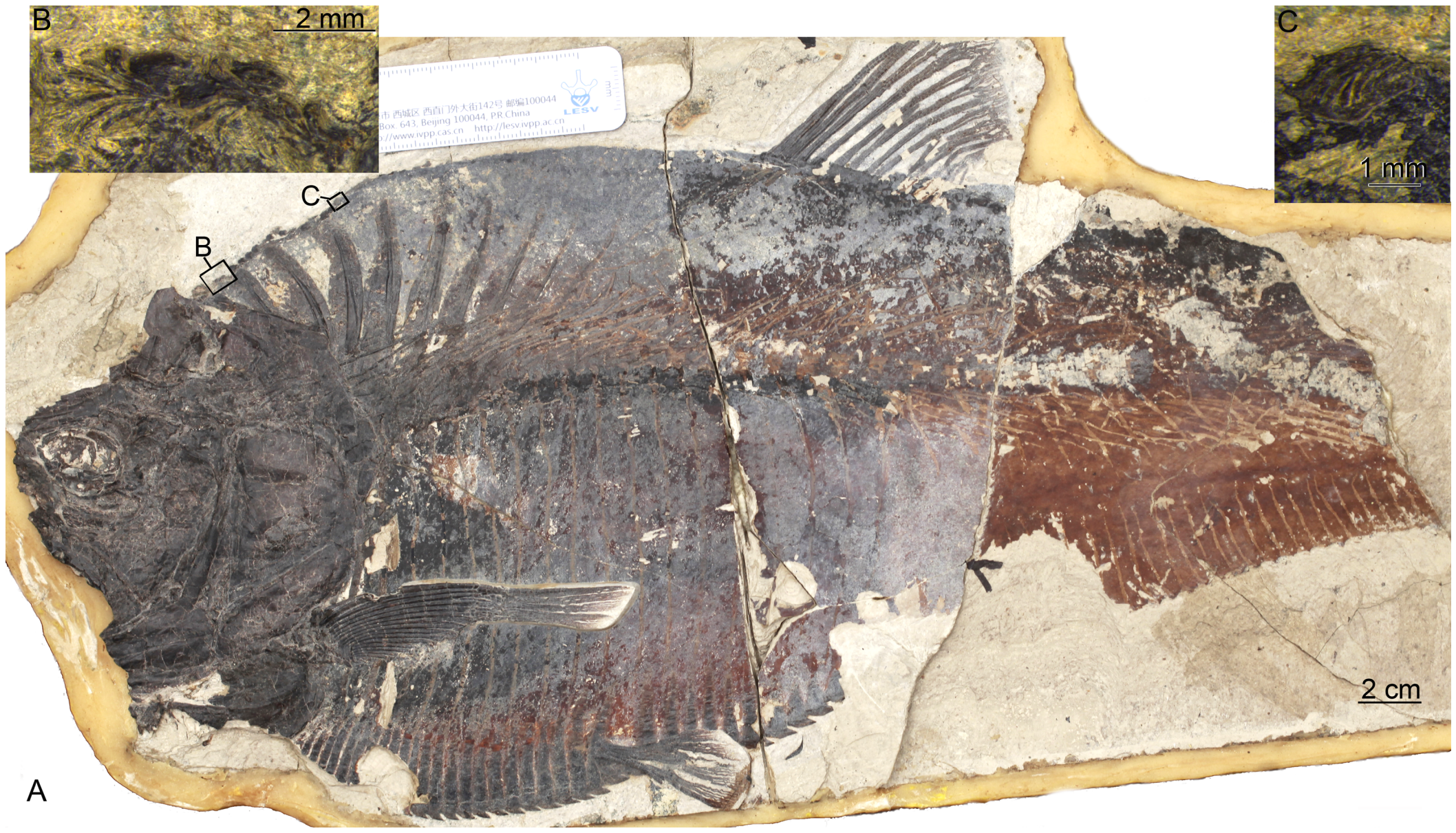

**Figure 6** ***Guiclupea superstes*, gen. et sp. nov.** (A) photograph of NHMG 033658 (image credit: Qiongyao Fu), (B) and (C) close up of the predorsal scutes in the black boxes of (B) and (C) in (A), respectively (image credit: Xueqiang Lei).

(3) **Opercular Series and Cheek Bones**. The opercle is trapezoidal in shape. Its ventral part is wider than the dorsal part, with the anteroventral corner protrudes downward and a little bit forward. The depth of the opercle is about 1.5 times of its width. No ornamentation on the surface of the opercle can be observed (Figs. 4A, 6). Two arms of the preopercle, with the dorsal branch slightly longer than the ventral branch, form an obtuse angle. The preopercular sensory canal runs along the mid-line of the bone sending out several branches backwards and downwards. Interopercle and subopercle are long and thin, with smooth surface (Figs. 4A–4B, 6A). About 8–9 branchiostegal rays can be detected in the holotype, although the outline of each ray is not very clear (Fig. 2A). In NHMG 033659, five of the posterior branchiostegal rays of the right side can be counted below the interopercular bone, while four displaced, slender anterior branchiostegal rays are discernible in the position anterior to the interopercle (Figs. 4A–4B).

(4) **Circumorbital Bones**. There is an arched, long bone above the frontal in NHMG 033659 (Figs. 4A–4B), we suspect it ought to be the supraorbital bone displaced from its original position. The sclerotic ring, consisting of two halves, can be observed in the posterior and anterior part of the orbit. Anterior to the orbit, two bones seem to bear sensory canals. The large, sub-triangular, anterior thin bone is the lacrimal, whereas the posterior rectangular one may be infraorbital 2 (Figs. 4A–4B). Detached infraorbital bones are preserved in NHMG 033660 (Fig. 5).

(5) **Jaws and Palate**. The mouth is somewhat superterminal on NHMG 004929. The oblique gape is relatively short, with the lower jaw articulation under the anterior border of the orbit (Fig. 4A; NHMG 004929). The upper jaw consists of a premaxilla, maxilla,

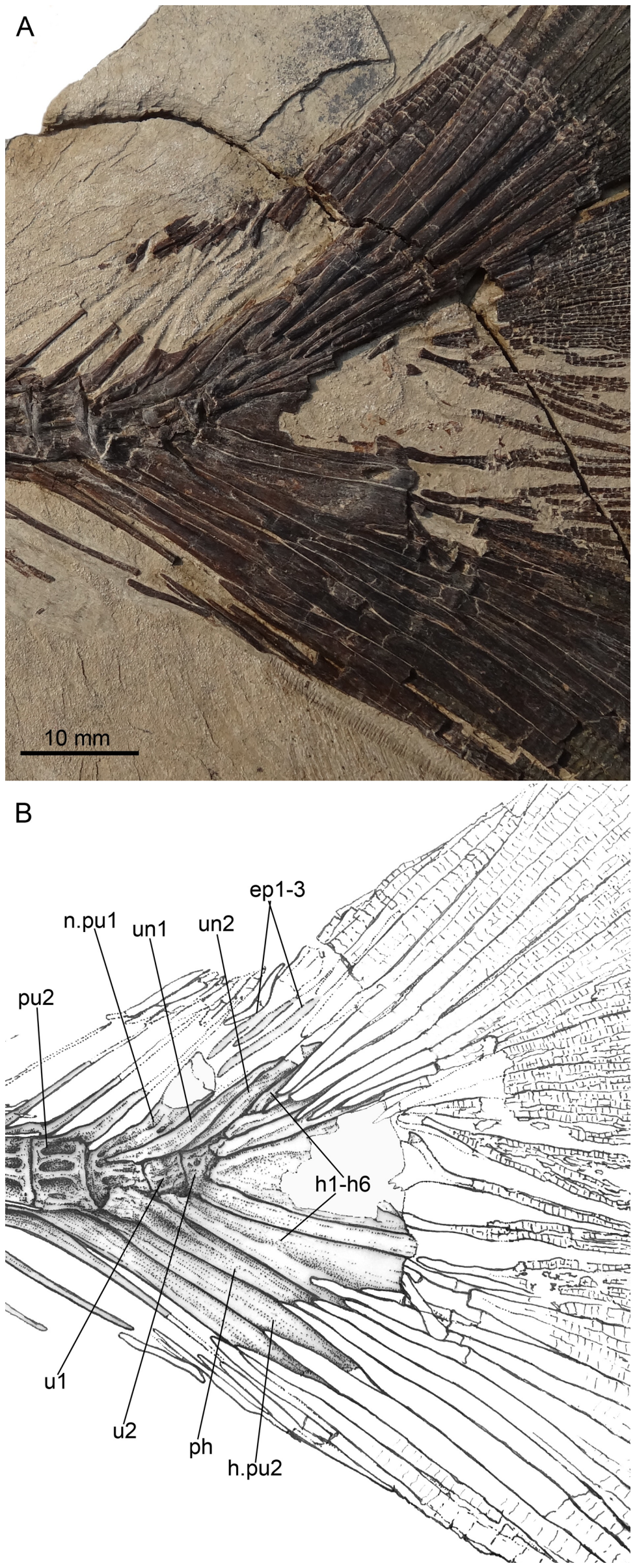

**Figure 7** ***Guiclupea superstes*, gen. et sp. nov.** (A) Photograph and (B) line drawing of the caudal skeleton NHMG 011648. Anterior facing left.

and two supramaxillae. The premaxilla is a small, long, and triangular bone, with a row of small conical teeth on its oral margin (Fig. 4C). The maxilla is a long bone, with its anterior part narrow and thick while its posterior part broadens into a thin blade and bears a rounded ventral profile. The oral margin of the thin blade is finely serrated. The anterior end of the maxilla develops into a round ethmoid head and a round palatine head (Fig. 5, NHMG 004929, 033684, 033686, 033689). Along the dorsal edge of the maxilla, two supramaxillae can be detected in NHMG 004929 and 033659 (Fig. 4A). In NHMG 033689, two disarticulated supramaxillae bones seem to be roughly equal in size and similar in

**Table 1** **Measurements and counts for the specimens of the *Guiclupea superstes*, gen. et sp. nov.** Measurements are in millimeters.

| | 004929 | 005532 | 033658 | 033659 |
|---|---|---|---|---|
| total length | | 526.0 | | 630+ |
| standard length (SL) | 502+* | 453.0 | 410+ | 585+ |
| head length | 175 | 132.0 | | 140+ |
| head length/SL | | 29.1% | | |
| head depth | 169.0 | 116.0 | | 165+ |
| body depth | 213 | 137.0 | 150 | 210 |
| body depth/SL | | 30.2% | | |
| predorsal length | 333.5 | 252.8 | | 310+ |
| predorsal length/SL | | 55.8% | | |
| prepelvic length | 293.7 | 242.0 | | 260+ |
| prepelvic length/SL | | 53.4% | | |
| preanal length | 387.0 | 309.3 | | 358+ |
| Preanal length/SL | | 68.3% | | |
| dorsal fin rays | | i, 12 | at least 13 | |
| anal fin rays | | ∼35 | | |
| pterygophores of anal fin | | ∼36 | 29+ | 22+ |
| pectoral fin rays | | at least 12 | 18 | 12+ |
| pelvic fin rays | | 5∼6 | 5∼6 | |
| abdominal vertebrae | 20 | 22 | 20 | 20 |
| caudal vertebrae (exclude u1, u2) | 12+ | 23 | | 24 |
| total vertebrae (exclude u1, u2) | 32+ | 45 | | 44 |
| predorsal bones | 10 | 10 or 11 | 10 | 10 |
| abdominal scutes | at least 36 | ∼38 | 35+ | 27+ |
| pre-pelvic scutes | ∼24 | ∼24 | ∼24 | 16+ |
| post-pelvic scutes | at least 12 | 14 | 11+ | 11+ |
| pre-dorsal scutes | | 55 | 27+ | |
| pairs of ribs | 18 | 20 | 18 | 19 |

**Notes.**
"+" stands for the actual digital larger than this digital because of the specimen is incomplete or not well-preserved.

shape; and their external surfaces are smooth, except for a low ridge extending along their midline.

The mandible has a well-developed coronoid process formed by the dentary and the anguloarticular. In NHMG 033681-3 and NHMG 038777, there is a single row of small conical teeth along the short oral margin of the dentary (Figs. 4D–4E). Teeth close to the symphysis of the two dentaries are slightly stouter than those in the rear; but in specimen NHMG 033660, no teeth can be seen on the oral margin of the dentary. These may have been lost during preparation or fossilization. Along the lower lateral margin of the dentary, the mandibular sensory canal is well developed with 6∼7 pores (Fig. 5). The angulo-articular is a triangular bone with the mandibular sensory canal running along its lower margin of the lateral surface. The length of angulo-articular is about half that of the dentary. Its posterior end forms the articular facet for the quadrate (Figs. 4A–4B, 5). The very small

retroarticular bone is located below the postarticular facet of the angulo-articular bone (Figs. 4A–4B).

The quadrate, as is generally for teleosts, consists of a fan-like plate at its dorsal side and a rod-like posterior process at its ventral side. Its articular head fits into the socket at the postero-dorsal end of the angulo-articular (Figs. 4A–4B). The parasphenoid can be partly observed in NHMG 033658 and 004929. It is difficult to judge whether or not a basipterygoid process and the "osteoglossid" tooth patch of the bone in most basal teleosts is present because of the preservation.

(6) **Hyoid Arch and Gill Arches**. The hyomandibular is a hatchet-like bone. It bears a thin, broad antero-dorsal plate and a long, narrow ventral shaft, which ventrally connects to the upper end of the symplectic (Figs. 4A–4B, 5). The condyle for articulation with the opercle is large. From that level, a prominent ridge runs ventrally along the posterior margin of the outer surface of the shaft. The foramen for hyomandibular branch of facial nerve (VII.hm) is clear. The symplectic inserts into the notch between the plate of the quadrate and its ventral process (Figs. 4A–4B). The detached entopterygoid is visible in NHMG 033660 and 033661. It is a broad, triangular bone, with numerous fine conical teeth covering its buccal side (Figs. 4F, 5). The metapterygoid is an expansive, trapezoid bone. Its anterior margin is posterior to the anterior margin of the quadrate, and its posterior margin reaches a relatively more dorsal position, almost in line with the hyomandibular condyle (Figs. 4A–4B, 6A).

The anterior ceratohyal is a thick, sub-rectangular plate with its length about twice of its depth. Its dorsal margin is slightly convex, whereas its ventral margin is slightly concave. Its central part is pierced by a large elongated oval foramen (Figs. 4F, 5) as in primitive clupeomorphs (*Grande, 1985*). The posterior ceratohyal is a triangle plate without a foramen within it. There is a small notch on its posterordorsal margin (Fig. 4G). The urohyal shows a narrow ventral keel and a vertical crest. The height of the crest gradually increases posteriorly (Fig. 5). Gill arches are not well-preserved, but many dislocated, long, pointed gill rakers with a bifid base that embraced the gill arches are observed in several specimens (Figs. 3A, 4F–4G, 5). The length of the gill rakers varies from about one vertebral centrum to 2∼3 times as long as a vertebral centrum or even more. There are numerous fine conical denticles recurved posteriorly throughout almost the whole upper edge of the gill rakers, differing from the situation in *Diplomystus* sp. from the English chalk, in which the rakers appear to be smooth throughout most of their length (*Forey, 2004*)

(7) **Paired Fins and Girdles**. The posttemporal is a large bone (Figs. 4A–4B; NHMG 004929). The supracleithrum is small, lying below and posterior to the very well-developed posttemporal. The cleithrum is a long, S-shaped bone, with its upper end covered by the supracleithrum. Below the cleithrum is the well-developed, laminate coracoid with a large notch on its anterior margin (Figs. 4A, 5–6). The pectoral fin is located rather high on the flank. The fin is long, extending past the insertion of the pelvic fin in NHMG 033658 (Fig. 6A). In other specimens, the fin rays do not look as long, probably because the distal ends of the fin rays were missing during the process of fossilization. Eighteen pectoral fin rays can be counted (Fig. 6A).

The pelvic girdle cannot be observed because of the covering of the abdominal scutes. The pelvic fin is small, with about 5∼6 fin rays, inserted at the level in front of the origin of the dorsal fin (Figs. 2A, 6A). The length of the longest pelvic fin ray is equivalent to the span of 6∼7 postpelvic scutes.

(8) **Dorsal and Anal Fins**. The origin of the dorsal fin is situated slightly posterior to the mid-point of the standard body length. There are about 14 dorsal fin rays (Figs. 2A, 6A). The first two are short and unbranched, while the third through fifth rays are the longest. Twelve pterygiophores are preserved in the holotype. The first is comparatively long and broad, inserted between the tenth and eleventh neural spines, whereas those posterior to it are much narrower (Fig. 2A).

The origin of the anal fin is more posterior than the end of the dorsal fin base, closer to the pelvic fin insertion than to the caudal fin base. The anal fin base is comparatively long, containing about 38 rays, of which the anterior six are longer than the posterior ones. At least 36 pterygiophores are preserved in the holotype. In the specimen NHMG 038778, 38 pterygiophores can be counted. Anterior pterygiophores are longer than the posterior ones. The first pterygiophore inserts between the last rib and the first hemal spine (Figs. 2A, 3A).

(9) **Vertebral Column**. Twenty-three caudal vertebrae, excluding two ural centra, and nineteen abdominal vertebrae are recognized in the holotype (Figs. 2A–2B). We added two to our counts for the vertebrae that normally lie under the superficial bones of the skull and pectoral girdle (e.g., opercle, cleithrum); thus, the total number of the preural vertebrae is about 44 in the holotype. The length and depth of the vertebra are about equal, except the last few, which are shorter than the anterior ones (Figs. 2A, 3A). There are two longitudinal ridges along the lateral side of each vertebra, forming two pits on their lateral side (Fig. 3A). Halves of the neural arches are fused medially. The hemal spines start from the 21st or 22nd centrum, and their length decreases gradually until the fourth or fifth preural centrum where they increase greatly to support the fin rays of the lower caudal lobe (Fig. 7).

Nineteen pairs of ribs are present in the holotype, but in NHMG 033659 only 18 pairs of ribs could be counted. All the ribs insert deeply into the centra. Ventrally, these ribs touch the lateral wings of the abdominal scutes. There are numerous thin, long epineural and epipleural intermuscular bones. The epineural series extends from the occiput to the first preural centrum. The epipleural series starts from approximately under the last three abdominal vertebrae and extends to about the first preural centrum. The longest epineural reaches the length about five to six centra, and the longest epipleural is about the length of four to five centra (Figs. 2A–2B, 3A, 6).

(10) **Caudal Skeleton and Fin**. The caudal skeleton and fin are preserved relatively well in specimens NHMG 011646-011651. The neural and haemal spines of the second to the fourth or fifth preural centrum are elongated and somewhat flattened in that of the second and third preural centrum, and support a few caudal fin rays and procurrent rays. The structure of the caudal skeleton, as a whole, differs from that in clupeiforms but closely resembles that in ellimmichthyiforms, i.e., bearing at least two autogenous uroneurals, the first one not fusing with the first preural centrum as in Clupeoidei, although it is

long and thick, extending anteriorly to reach the anterodorsolateral side of that centrum. The second uroneural is much shorter than the first one, extending anteriorly only to the anterior end of the second ural centrum, although its distal end reaches that of the first one. It cannot be confirmed if a third uroneural is present or not. There are two free ural centra; the first one is about equal in size to the first preural centrun, but the second one is much shorter than the first one. Six hypurals are present. The proximal end of first hypural is in contact with, but not fused to, the first ural centrum, although the narrow second hypural is fused to the second ural centrum at its proximal end. Hypural 3 is the largest. Anteriorly, its enlarged proximal end contacts with the distal end of the second ural centrum entirely constraining the hypurals 4–6 reach anteriorly to contact the second ural centrum; posteriorly, it expands, filling the entire space between the second and the fourth hypurals, such that there is no diastema between them, as in *Armigatus brevissimus* and the Eocene *Diplomystus* species. The fourth through sixth hypurals become narrower and shorter. The proximal end of the parhypural is fused with the first preural centrum, which has a long, broad neural arch and spine. There are three epurals (NHMG 011650). The caudal fin is deeply forked with the upper and lower lobes of about equal length, containing 19 principal fin rays (I, 9-8, I), and eleven and nine procurrent rays above and below the principal caudal fin rays, respectively (Figs. 2A, 7). The proximal ends of the middle principal fin rays are preserved as impressions in NHMG 011648, but the outline of the lowermost ray end of the upper lobe can be detected, it is significantly enlarge. On NHMG 011649, the ends of two lowermost rays of the upper lobe and two uppermost rays of lower lobe enlarge obviously, but only the end of the lowermost ray of upper lobe enlarges significantly as in NHMG 011648. Three caudal scutes can be seen in the dorsal margin of the caudal fin in NHMG 011647.

(11) **Predorsal Bones and Scutes**. There are ten or eleven predorsal bones with thin anterior and posterior bony expansions. The anterior bones are broader than the posterior ones, and the first three stretch almost vertically (Figs. 2A–2B, 3A) or somewhat postero-ventrally (Fig. 6), whereas the rest are oriented antero-ventrally.

There is a series of scutes along the dorsal margin from the occiput to the origin of the dorsal fin in the holotype. The entire series includes about 55 small, equally-sized scutes (Fig. 2). Because of preservation, the details of the scutes cannot be observed in this specimen. In specimen NHMG 038778, about six predorsal scutes from immediately behind the occiput and about nine immediately anterior to origin of the dorsal fin can be recognized. The detail of the scutes cannot be observed also because of poor preservation. However, details can be discerned in NHMG 033658 (Fig. 6), in which numerous small dorsal scutes are preserved along the dorsal margin of the body from the occiput to the seventh predorsal bone. Most of the scutes are displaced, some of them are even turned upside-down and thus show their smooth ventral surface, but many scutes show their dorsal surface ornamented with several radial ridges. In NHMG 033659, a few predorsal scutes with weak ridges are preserved anterior to the dorsal fin. In NHMG 033680, several displaced dorsal scutes can be detected bearing radial ridges similar to those of NHMG 033658. No dorsal scutes are seen behind the dorsal fin base in any specimen.

About 24 prepelvic scutes are counted from the posterior edge of the coracoid to the insertion of the pelvic fin in the holotype (Figs. 2A–2B) and in NHMG 033658 (Fig. 6). Fourteen postpelvic scutes are present in the holotype. Only the first 11 postpelvic scutes are preserved in NHMG 033658. Several much smaller ventral scutes can be detected below the coracoids in the holotype or displaced in the lower part of the head in NHMG 033659. The scutes behind the coracoid bear a strong ventral spine and much higher lateral wings. The lateral wings are wider at their ventral edges, but narrow gradually dorsally, extending from the ventral edge of the body up to about one quarter of the way to the vertebral column (Figs. 2A, 3A, 6).

(12) **Squamation/Scales**. In NHMG 038778, the impression of some of the scales can be seen in the body above the vertebral column. The scales are small. Details are not clear.

# DISCUSSIONS

## Phylogenetic relationships of the new form

Although the predorsal scutes of the new form do not expand laterally as in the diagnosis given by *Grande (1982)* of the Ellimmichthyiformes when he established the order, the presence of the two parietals meeting at the midline, a beryciform foramen within the anterior ceratohyal, ornamentation on the predorsal scutes, and the structure of the caudal skeleton suggest that the new species differs from clupeiforms but, instead, resembles ellimmichthyiforms (*Grande, 1982*; *Grande, 1985*). To further assess the systematic position of the new form, phylogenetic analyses were conducted.

Two data matrices were constructed for the phylogenetic analyses. Data matrix 1 (D1) used *Chanos chanos* as an outgroup taxon, whereas data matrix 2 (D2) used *Elops saurus* instead of *Chanos chanos* as the outgroup taxon. Each data matrix includes 55 morphological characters and 40 taxa (see Appendices S2–S3), analyzed using parsimony and Bayesian inference methods respectively.

The analysis of D1, using parsimony criteria, generated four equally most parsimonious trees (MPTs). A strict consensus tree (SCT1) of 189 steps was built, with a consistency index (CI) of 0.323 and retention index (RI) of 0.670 (Fig. 8, Appendix S1). The SCT1 shows two main clades of Clupeomorpha: Clupeiformes and Ellimmichthyiformes. The monophyly of the Ellimmichthyiformes, including *Ornategulum* as the most basal taxon, is supported by the following features: two parietals meeting at the midline (2:0), two supramaxillary bones (8:0), presence of the "basipterygoid" process (9:1), anterior ceratohyal with foramen (11:1), and three epurals (38:0). Three of these four characters (i.e., 2, 9, and 11) are uniquely derived characters (ci = 1) for this clade. *Gasteroclupea* and *Sorbinichthys* lie in the basal position of the Ellimmichthyiformes, but do not form sister groups as suggested by *Marramà et al. (2019)* and *Boukhalfa et al. (2019)*. Sorbinichthyidae *sensu* (*Murray & Wilson, 2013*) including only the two *Sorbinichthys* species, is strongly supported by a number of derived characters: broad dorsal process of the posttemporal (15:2), posterior predorsal scutes laterally expanded (41:1), the most posterior predorsal scutes enlarged (44:1), high number of abdominal scutes (51:2), but fewer postpelvic scutes (52:1). Monophyly of *Armigatus* is supported by sharp proximal end of first hypural (27:1), predorsal scute series incomplete

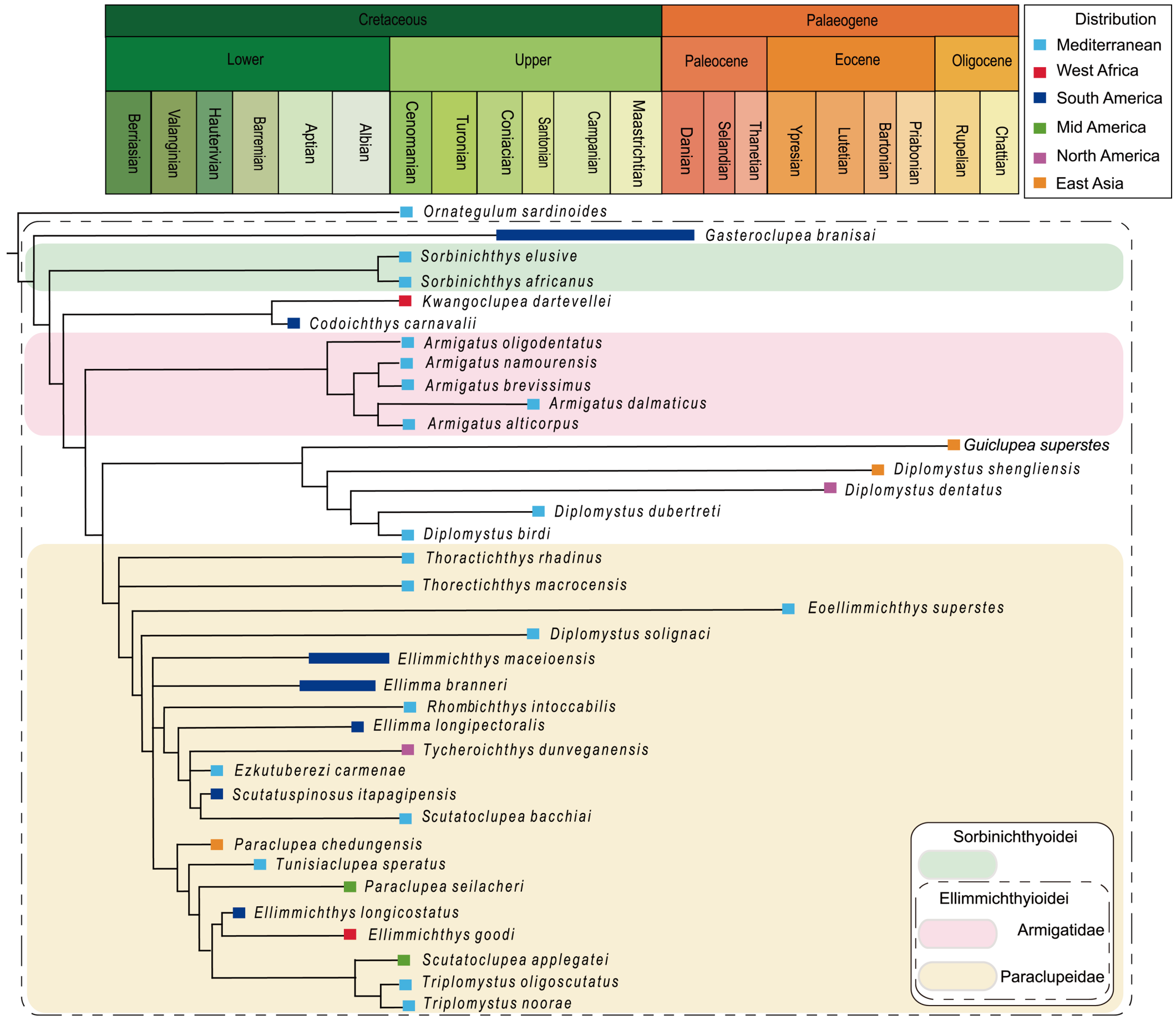

**Figure 8 Strict consensus of most parsimonious trees.** Strict consensus of most parsimonious trees retrieved in TNT 1.5 based on 55 morphological characters and 40 taxa, with *Chanos chanos* being used as outgroup taxa.

(39:0) (ci = 1). Monophyly of *Diplomystus* (excluding *D. solignaci* Gaudant & Gaudant, 1971) is supported by the presence of sub-rectangular scutes in anterior and posterior predorsal series (40:1 and 41:1), and presence of a series of spines on posterior margin of lateral wings of predorsal scutes (42:1) (ci = 1). *Diplomystus solignaci* is a member of the paraclupeid clade as suggested in many previous studies (Alvarado-Ortega, Ovalles-Damián & Arratia, 2008; Murray & Wilson, 2013; Figueiredo & Ribeiro, 2016; Marramà et al., 2019; Boukhalfa et al., 2019). *Armigatus* is in a more basal position of *Diplomystus* as suggested by Alvarado-Ortega, Ovalles-Damián & Arratia(2008) and Figueiredo & Ribeiro (2016). Our new form, *Guiclupea*, forms the sister group to *Diplomystus* sensu stricto. The membership of Paraclupeidae, not including *Kwangoclupea* and *Codoichthys* as some previous studies suggested (e.g., Murray & Wilson, 2013; Figueiredo & Ribeiro, 2016; Marramà et al., 2019; Boukhalfa et al., 2019) but in some ways consistent with the recent analysis conducted by Vernygora & Murray (2020), is supported by the dorsal outline forming a marked angle

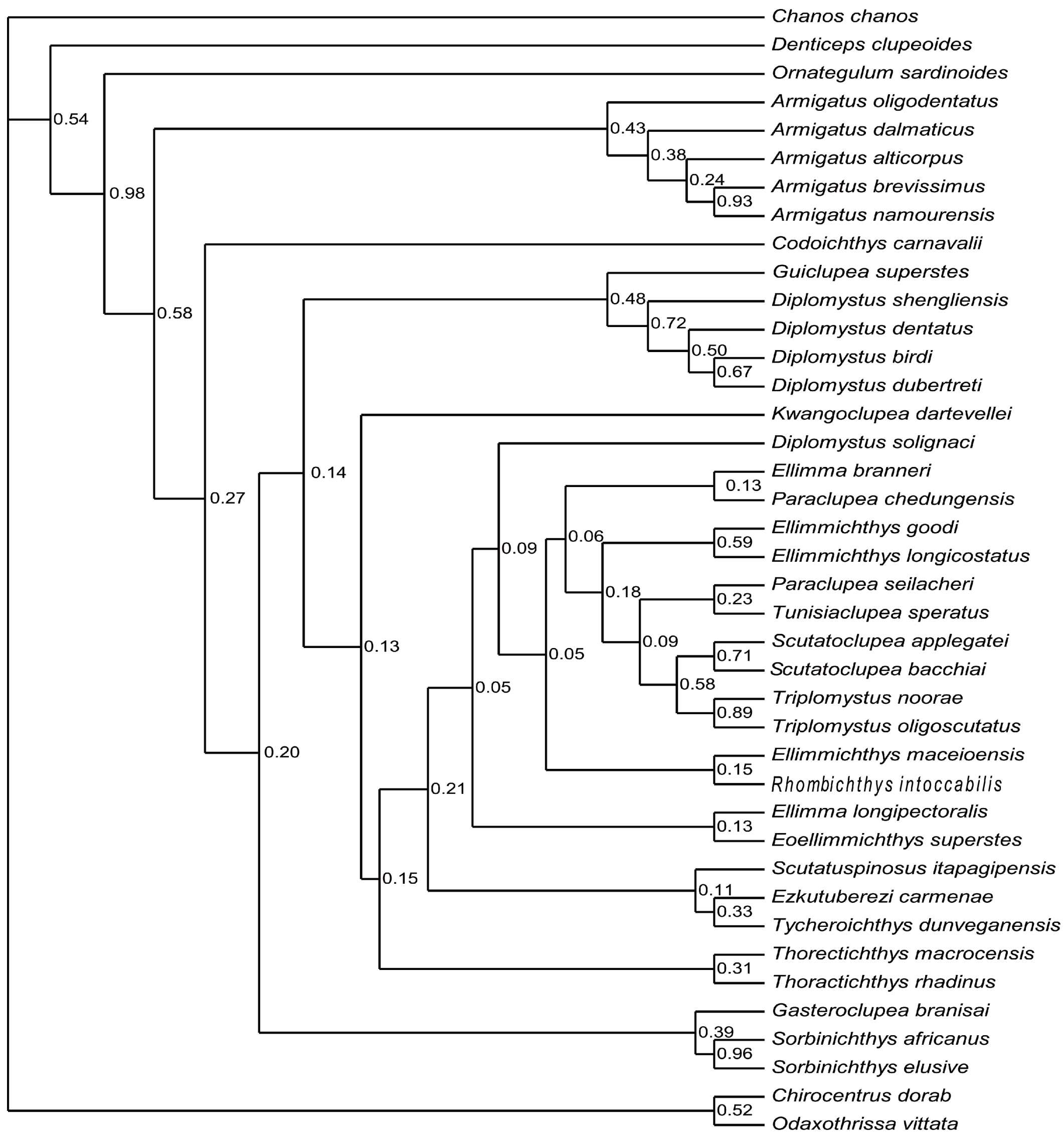

**Figure 9 Cladogram.** Cladogram resulting from Bayesian phylogenetic analyses based on 55 morphological characters and 40 taxa, with *Chanos chanos* used as outgroup taxon. The numbers at the internal nodes are the posterior probabilities of the corresponding clades.

at dorsal-fin insertion (1:1), first uroneural extending forward to second preural centrum (32:0), and predorsal scutes with ridges on the dorsal surface (44:1).

The analysis of D1 using a Bayesian inference method, generated a Bayesian Inference tree (BIT) (Fig. 9). As in SCT1, the monophyly of *Sorbinichthys*, *Armigatus*, *Diplomystus* sensu stricto, and Paraclupeidae is supported, and *Guiclupea* is sister to *Diplomystus* sensu stricto. Unfortunately, the monophyly of the Clupeiformes cannot be supported, and *Gasteroclupea*, *Sorbinichthys*, and *Kwangoclupea* lie in a more derived position than that of SCT1.

The analysis of D2, using parsimony criteria, recovered 16 MPTs. The SCT of the 16 MPTs (see Appendix Fig. 3S) is 188 in step, the CI is 0.324, and the RI is 0.671. The cladogram of SCT2 shows that *Ornategulum* is in the basalmost position of the superorder, i.e., *Ornategulum* does not belong to the Ellimmichthyiformes as suggested by SCT1 and BIT1. The monophyly of the Clupeiformes, *Sorbinichthys*, and the remaining members of the Ellimmichthyiformes clade exclusive of *Gasteroclupea* was all supported, and the last one has the same topology as in SCT1.

The topology of BIT2, resulting from the analysis of D2, is very similar to that of BIT1 except for the position of *Ornategulum* and *Denticeps.* In BIT2, *Ornategulum* lies in the most basal position as in SCT2, and *Denticeps* is sister group to the two clupeoid genera. It seems that applying different outgroup taxa affects the position or the assignment of *Ornategulum*, as demonstrated by the previous analyses (*Murray & Wilson, 2013*; *Marramà et al., 2019*; Figuereido & Ribeiro 2017; *Boukhalfa et al., 2019*). In addition, the positions of *Armigatus*, *Gasteroclupea*, *Sorbinichthys*, *Kwangoclupea*, and *Codoichthys* remain the same in the two SCTs and in the two BITs respectively, but differ between the SCTs and the BITs. In the BITs, *Armigatus* lies in the basalmost position of the order but it is not so in the SCTs. In the BITs, *Gasteroclupea* belongs to the Sorbinichthyidae as suggested by *Marramà et al. (2019)* and *Boukhalfa et al. (2019)*; but in the SCTs, *Gasteroclupea* does not form the sister group to *Sorbinichthys. Kwangoclupea* forms the sister group to *Codoichthys* and lies in a relatively basal position in the two SCTs, whereas it lies in a relatively derived position and belongs to the Paraclupeidae in the two BITs. These differences between the general topologies of the SCTs and BITs are probably due to the fact that the information contained in the dataset is insufficient to draw firm conclusions about their relationships as pointed out in the recent analyses of the phylogeny of Ceratomorpha (*Bai et al., 2020*). To improve the understanding of relationships of the group, more phylogenetically informative fossils and more complete data are needed.

Although there are discrepancies between the MPTs and BITs, and between the trees with alternative outgroups, the general topologies of the four trees mentioned above are basically similar, and all the four trees suggest that the new form is a member of the ellimmichthyiforms and forms a sister group to *Diplomystus* sensu stricto. The close relationship of these two is supported by high supraoccipital crest (4:1), pelvic-fin insertion in advance of dorsal fin origin (22:0), and number of predorsal scutes ≥20 (46:1). Actually, in addition to these synapomorphies, the new form and *Diplomystus* sensu stricto, especially the Eocene species, i.e., *D. dentatus* and *D. shengliensis*, share many more similar characters, such as having an elongated fusiform body form, dorsal outline curved gently, no ornamentation on the skull bones, entopterygoid with teeth, high number of anal fin rays (23–25 in *D. birdi*, 27 in *D. dubertreti*, 38–41 in *D. dentatus*, about 39 in *D. shengliensis*, and about 38 in *Guiclupea superstes*), close-to fan shape arrangement of the predorsal bones, and no diastema between the second and third hypural (but there is a gap between second and third hypural in the Late Cretaceous species, i.e., *D. birdi* and *D. dubertreti* (*Chang & Maisey, 2003*, p27). The last character also occurs in pristigasteroids, and osteoglossids, some elopomorphs, and a number of ostariophysans (*Chang & Maisey, 2003*). The differentiation between *Diplomystus* and *Guiclupea* is in the

shape and ornamentation of the predorsal scutes (sub-rectangular vs. ovate, presence vs. absence of pectinate posterior border, dorsal surface smooth vs. with radial ridges), and the number of predorsal bones (6–8 vs. 10–11). Accordingly, the new form is a distinct genus and species, and can be easily distinguished from *Diplomystus*. Comb-like teeth along the posterior edges of the dorsal scutes is a derived character of *Diplomystus*. In addition to *Diplomystus*, sub-rectangular predorsal scutes also occur in most members of the Paraclupeidae. As far as the radial ridges on the dorsal surface of predorsal scutes is concerned, they usually occur in paraclupeids, such as *Paraclupea*, *Ellimmichthys*, *Ellimma*, *Triplomystus*, etc., and these forms usually have a marked angle at the insertion of the dorsal fin, and sub-rectangular dorsal scutes at least in the posterior part of the scute series. The new form is distinct from them in the shape of the body and predorsal scutes. Among the species with ornamentation on the predorsal scutes, the new form resembles *Scutatuspinosus itapagipensis* in the shape of the body and predorsal scutes (not laterally expanded), and posterior expansion of the third hypural, leaving no gap or notch between the second and third hypurals. However, there are obvious differences between the two forms in the number of predorsal scutes and anal fin rays, ornamentation on the skull roof bones, and the size and shape of the abdominal scutes. Previous studies (*Yabumoto, 1995*; *Chang & Grande, 1997*) suggest that "*Diplomystus*" from Japan (*Uyeno, 1979*; *Yabumoto, Yang & Kim, 2006*) are closely related to *Paraclupea chetungensis* in having ridges on the dorsal scutes, however, the shape of the body (not deep, no marked angle at the origin of the dorsal fin) and dorsal scutes (not laterally expanded, all scutes about the same size) of the Japan material are obviously different from that of *Paraclupea* but resemble that of the new form, but the new form differs from the "*Diplomystus*" from Japan at least in the neural spines of the vertebrae not seperated and the number of anal fin rays, vertebrae, predorsal and ventral scutes, and predorsal bones. To assess the systematic position of the "*Diplomystus*" from Japan, reexamination of the material is needed. On the whole, the new form displays a mosaic combination of characters. It bears radial ridges on the dorsal surface of the predorsal scutes as in the paraclupeids, but the scutes are all about the same size as in *Armigatus* and *Diplomystus*. Consequently, *Guiclupea* can easily be distinguished from all known ellimmichthyiforms in number and morphology of the dorsal scutes and allow us to recognize that predorsal scutes with ridges on the dorsal surface are not unique to Paraclupeidae.

## Body shape and size of the Ellimmichthyiformes

The ellimmichthyiforms are diverse in both general morphology and body size. Generally, ellimmichthyiforms show two types of body form. One bears a deep body, with the maximum depth/standard length (MD/SL) larger than 50%, and some of them even with the MD roughly equal to, or slightly larger than the SL in adult specimens, e.g., in *Tycheroichthys dunveganensis* and *Rhombichthys intoccabilis* (*Hay et al., 2007*; *Khalloufi, Zaragüeta-Bagils & Lelièvre, 2010*). Most of them are referred to the paraclupeids. The other kind of fishes have an elongate fusiform shape, the MD/STL often less than 50%. They occupy a relatively basal position in the Ellimmichthyiformes, such as *Armigatus*, *Diplomystus*, and *Guiclupea*.

Body size of ellimmichthyiforms ranges from several centimeters to about 65 cm in total length (TL). The known smallest fish is *Eoellimmichthys superstes* with the TL and SL about 17.5 mm and 13.7 mm, respectively (*Marramà et al., 2019*). Many species have a TL less than 100 mm, such as *Armigatus alticorpus*, *A. dalmaticus*, *A. oligodentatus*, *Codoichthys carnavalii*, *Diplomystus shengliensis*, "*Diplomystus*" *trebicianensis*, *Ellimmichthys maceioensis*, *Gasteroclupea branisai*, *Scutatuspinosus itapagipensis*, *Sorbinichthys africanus*, *Thorectichthys marocensis*, *T. rhadinus*, and *Tunisiaclupea speratus*. Species with TL larger than 200 mm are rare. The TL of *Rhombichthys intoccabilis* reaches about 230 mm in the holotype (*Khalloufi, Zaragüeta-Bagils & Lelièvre, 2010*, Fig. 3). The predorsal length of *Horseshoeichthys armigserratus*, from the Maastrichtian of Canada, is 172 mm, with the estimated SL about 260–280 mm (*Newbrey et al., 2010*). Specimens with SL over 300 mm are only seen in *Diplomystus dentatus* from the Eocene of the United States, on the eastern side of the Pacific and *Guiclupea superstes* from the Oligocene of South China, on the western side of the Pacific so far. The former reaches a TL of about 650 mm (*Grande, 1982*) while the latter reaches a SL about 600 mm. They are the largest ellimmichthyiform fishes known. It is noted that the phylogeny of the *Diplomystus* clade show a trend toward increase their body size, as the trend has been observed in many clades of ray-finned fish (*Guinot & Cavin, 2018*) over a long time interval. In the Cretaceous, members of this order usually have a small body length; it is not until the end of the Cretaceous that some members (e.g., *Horseshoeichthys armigserratus*) attained a relatively large body-size; in the late Paleogene, some members developed an even larger body size. It is worth mentioning that the fishes with large body size all are members of the *Diplomystus* clade (*Horseshoeichthys* forms the sister group to *Diplomystus* species (Veryngora & Murray, 2020)) and occur on the sides of the Pacific. Both *Horseshoeichthys armigserratus* from the end-Cretaceous and *Guiclupea superstes* from the Oligocene occurred at a time of global cooling, the former even lived at about 60° N paleolatitude (*Newbrey et al., 2010*), the latter lived in the environment where the temperature lower than the 22 °C of the Ningming area today (*Shi, Zhou & Xie, 2012*). However, the *Diplomystus dentatus* occurred in early-middle Eocene, at a time of global greenhouse climate or the early Eocene Climatic Optimum *Zachos et al. (2001)*, and no large-size of the conterporary *Diplomystus shengliensis* and *Eoellimmichthys* have been reported. So, the relationship of the body size with the climate within this group is no clear.

Several studies indicated that enlarged body size of fishes are associated with their lifestyles. A recent study based on a comparative analysis indicates that across the Clupeiformes diadromous species are larger than non-diadromous species, for increased body size is an adaptation to the energetic long-distance migration. No association of body size with trophic position was found (*Bloom, Burns & Schriever, 2018*). Another study based on over 4500 migratory and non-migratory species of ray-finned fishes, also shows that migratory species are larger than non-migratory relatives in nearly all clades and across all modes of migration (*Burns & Bloom, 2020*). Based on extant and fossil data covering the Late Jurassic-Paleocene interval, *Guinot & Cavin (2018)* suggested that the proportion of body size shifts associated with environmental transitions more than within a given environment, especially for major positive body size shifts towards mixed environments.

From these studies, we have reason to speculate that the wide-spread *Diplomystus* clade possibly involve some amphidromous or migratory species.

## Paleobiogeographic history of the Ellimmichthyiformes

The Ellimmichthyiformes, like its sister-group the Clupeiformes ( *Lavoue et al., 2013*; *Lavoué, Konstantinidis & Chen, 2014*) is a cosmopolitan group of fishes, with members distributed worldwide in marine, euryhaline and freshwater, and exhibit a complex paleobiogeographic history (Fig. 1). The oldest known ellimmichthyiform fish so far is *Ezkutuberezi carmenae* from the Valanginian-Barremian of northern Spain (*Poyato-Ariza, López-Horgue & García-Garmilla, 2000*). Fossils from Hauterivian-Barremian include *Scutatuspinosus itapagipensis* and *Ellimmichthys longicostatus* from northeastern Brazil (*Figueiredo & Ribeiro, 2017*; *Cope, 1886*), *Tunisiaclupea speratus* from southern Tunisia (*Boukhalfa et al., 2019*), and *Paraclupea chedungensis* from eastern China (*Sun, 1956*; *Chang & Grande, 1997*; *Hu et al., 2017*). All these species belong to the Paraclupeidae and distributed in non-marine sediments. The paraclupeid fishes are abundant during the late Early Cretaceous (Aptian-Albian). There are *Ellimma branneri* and *Ellimmichthys maceioensis* from Alagoas (*Schaeffer, 1947*; *Chang & Maisey, 2003*; *Malabarba et al., 2004*), and *Ellimma longipectoralis* from Santos Basin (*Polck et al., 2020*) of Brazil; *Ellimmichthys goodi* from Equatorial Guinea (*Eastman, 1912*); and *Paraclupea seilacheri* from Puebla, Mexico (*Alvarado-Ortega & Melgarejo-Damián, 2017*). In addition to paraclupeids, there are *Codoichthys carnavalii* from the Aptian of Brazil (*Figueiredo & Ribeiro, 2016*), *Foreyclupea loonensis* from the Albian of Canada (*Vernygora, Murray & Wilson, 2016*, the authors thought this species should be closely related with *Scutatuspinosus itapagipensis*), and the recently described *Armigatus carrenoae* from marine Albian of Central Mexico (*Alvarado-Ortega, Than-Marchese & Melgarejo-Damián, 2020*). These indicate that the Ellimmichthyiforms had been diversified and distributed widely during the Early Cretaceous in Europe, South and North America, Africa, and East Asia. The close relationship between the Early Cretaceous fish faunas from northeastern South America and from western Africa might have resulted from the contiguous margins of Brazil and West Africa during the Early Cretaceous (*Chang & Grande, 1997*) or resulted from Tethys Sea and the South Atlantic Ocean were intermittently connected through North-South Trans-Saharan seaways, as postulated by *Lavoue et al. (2013)*. But there is little geologic evidence to support an Early Cretaceous non-marine paleogeographic connection between the eastern Asiatic margin and western Gondwana. Consequently, the distribution pattern of paraclupeids in the Early Cretaceous is faced with a biogeographic conundrum. No favorable hypothesis adequately explains this distribution pattern to date. *Chang & Maisey (2003)* suggested that either a substantial portion of their non-marine fossil record is missing or their distribution involved marine dispersal.

During the early Late Cretaceous, the Ellimmichthyiformes reaches their greatest diversity. Not only the Paraclupeidae and Armigatidae or *Armigatus* are highly diversified in the Mediterranean region and the former extended their range to North America (*Tycheroichthys dunveganensis* from Canada, *Hay et al., 2007*), but all other main ellimmichthyiform clades, i.e., Sorbinichthydae and *Diplomystus* clade, occurred and

flourished in the Cenomanian with their oldest record from the eastern Tethys (Lebanon) (*Woodward, 1895*; *Signeux, 1951*; *Grande, 1982*; *Zhang, Zhou & Qing, 1985*; *Bannikov & Bacchia, 2000*; *Murray & Wilson, 2011*; *Murray & Wilson, 2013*; *Murray et al., 2016*). With all main clades first occurring there, undoubtedly, the circum-Mediterranean region is a hotspot on the evolution of ellimmichthyiforms. Species of *Diplomystus* are also found from the Cenomanian English chalk (*Forey, 2004*). The diversity of ellimmichthyiforms during the early Late Cretaceous was probably resulted from the high sea surface temperatures, eustasy and the consequent land-sea distribution that increased food input, dispersal routes, and habitat fragmentation for these fishes (*Guinot & Cavin, 2016*; *Boukhalfa et al., 2019*). Toward the end of the Cretaceous, the diversity of the Ellimmichthyiformes suddenly declined. Only *Gasteroclupea branisai* from South America and *Horseshoeichthys armigserratus* from North America are known to date.

Rare ellimmichthyiforms survived after the Cretaceous-Paleogene boundary. *Gasteroclupea branisai*, *Eoellimmichthys superstes*, and *Diplomystus* clade are the exceptions. *Gasteroclupea branisai* first occurred in the Late Cretaceous of South America and survived to the Danian of Argentina and Bolivia (*Signeux, 1964*; *Marramà & Carnevale, 2017*). *Eoellimmichthys superstes* is a paraclupeid from the marine Eocene of Italy. Interestingly, *Diplomystus* clade not only survived up to the Oligocene, but also had a relatively wide distribution range along both sides of the Pacific Ocean (see Fig. 1). *Horseshoeichthys armigserratus* occurred in western side of North America at the end of the Cretaceous (*Newbrey et al., 2010*). The Eocene *Diplomystus* occurred on both sides of the Pacific Ocean (along the coast of the Bohai Gulf, east China, and in western North America) with species bearing a striking similarity in morphology (*Zhang, Zhou & Qing, 1985*; *Chang & Maisey, 2003*). Their sister group, *Guiclupea superstes*, survived to the Oligocene as the youngest ellimmichthyiform fish. The Eocene "transpacific" distribution pattern of *Diplomystus* and other fishes and terrestrial vertebrates has long been noted by paleontologists (*Chang & Chow, 1978*; *Grande, 1982*; *Grande, 1985*; *Zhang, Zhou & Qing, 1985*). A broad connection between Asia and North America in the Bering Strait area and temporary desalination of the Arctic Ocean could have facilitated the dispersal of these fishes (*Chang & Maisey, 2003*). It is interesting to find that all ellimmichthyiform fossil localities (Fig. 1) are close to the recent coast except some of the North American localities. Many contemporaneous fish faunas have been found from inland areas of China, Mongolia, and East Kazakhstan (*Tang, 1959*; *Liu, Liu & X., 1962*; *Wang, Li & Wang, 1981*; *Sytchevskaya, 1986*), but no ellimmichthyiforms have ever been reported from there yet. Many Cretaceous fishes are found from Zhejiang and Fujian Provinces, East China, but *Paraclupea* only occurred in the eastern region of the provinces, although there are many contemporaneous fishes, such as *Paralycoptera* and others, from the middle and western part of the provinces (*Chang & Chow, 1977*). In addition, most Recent Clupeomorphs inhabit the sea, a few dwell not far away from coastal regions; many taxa within a lineage co-occur in fresh and marine waters and in temperate and tropical areas, and water temperature and salinity are seen as the usual case and poor dispersal barriers for this group of fishes (*Lavoué, Konstantinidis & Chen, 2014*). It seems reasonable to suggest that the origin and dispersal of ellimmichthyiforms may have something to do with the sea.

## CONCLUSION

*Guiclupea superstes* from the Oligocene of south China represents the youngest record of ellimmichthyiform. Its occurrence indicates that the Ellimmichthyiformes had a wider distribution range and a longer evolutionary history than previously knew. *Guiclupea superstes* is closely related to *Diplomystus* sensu stricto, which suggests that the dorsal scutes with ridges on dorsal surface is not a character unique to paraclupeids.

Paraclupeids has a comparatively long evolutionary history. They were very diverse and widely distributed during the Early through early Late Cretaceous, and survived to the Eocene in the circum-Mediterranean area. *Sorbinichthys* and *Armigatus* were restricted to the Mediterranean region in the early Late Cretaceous, but *Armigatus* had been distributed in Mid America (Mexico) during Albian. *Diplomystus* clade might have originated not later than the Cenomanian, and disappeared from Europe and the Middle East after the early Late Cretaceous, but was still prospering in the Pacific sides from the end of the Cretaceous to Eocene and survived until the Oligocene. This allowed the Ellimmichthyiformes to obtain a particularly long distribution range on the western side of the Pacific.

There is still no consensus among ichthyologists on the phylogenetic relationships of the Ellimmichthyiformes, especially the relationship of *Armigatus* and *Diplomystus*, as well as the position of *Gasteroclupea* and *Codoichthys*. Besides, there are discordances between the most parsimonious tree and the Bayesian Inference tree. To solve these problems, more informative specimens and characters are needed to enhance the dataset.

### Institutional Abbreviations

| | |
|---|---|
| **IVPP** | Institute of Vertebrate Paleontology and Paleoanthropology, Chinese Academy of Sciences, Beijing, China |
| **NHMG** | Natural History Museum of Guangxi Zhuang Autonomous Region, China |

### Anatomical Abbreviations

| | |
|---|---|
| **aa** | angulo-articular |
| **ach** | anterior ceratohyal |
| **as** | autosphenotic |
| **br** | branchiostegal rays |
| **cl** | cleithrum |
| **cor** | coracoid |
| **cs** | caudal scute |
| **d** | dentary |
| **ect** | ectopterygoid |
| **en** | epineural |
| **enpt** | entopterygoid |
| **ep** | epural |
| **epl** | epipleural |
| **fr** | frontal |
| **h** | hypural |
| **hy** | hyomandibula |

| | |
|---|---|
| **io** | infraorbital |
| **iop** | interopercle |
| **la** | lacrimal |
| **le** | lateral ethmoid |
| **met** | metapterygoid |
| **ms** | mesethmoid |
| **msc** | mandibular sensory canal |
| **mx** | maxilla |
| **ns** | neural spine |
| **op** | opercle |
| **pa** | parietal |
| **pch** | posterior ceratohyal |
| **pcl** | postcleithrum |
| **pd** | predorsal bones (supraneurals) |
| **ph** | parhypural |
| **pmx** | premaxilla |
| **pop** | preopercle |
| **pr** | pleural rib |
| **ps** | parapophysis |
| **pt** | posttemporal |
| **pto** | pterotic |
| **pu** | preural centrum |
| **q** | quadrate |
| **ra** | retroarticular |
| **sc** | scapula |
| **scl** | supracleithrum |
| **scr** | sclerotic bone |
| **so** | supraorbital |
| **sop** | subopercle |
| **smxa** | anterior supramaxilla |
| **smxp** | posterior supramaxilla |
| **sp** | sphenotic |
| **spo** | supraoccipital |
| **sy** | symplectic |
| **u** | ural centrum |
| **uh** | urohyal |
| **un** | uroneural |
| **vo** | vomer |
| **vs** | ventral scutes |

## ACKNOWLEDGEMENTS

We thank Zhanxiang Qiu from IVPP for his kind help in creating the species name, Alison M. Murray from University of Alberta for providing references, Weicai Chen from Nanning Normal University for help in conducting MrBayes analyses, Desui Miao from University of Kansas for stylistic improvement. Thanks are extended to Lidi Cen, Xueqiang Lei, and Qiongyao Fu from NHMG for preparing the fossil specimens, Guodun Kuang, Qiuping Zhu, Zhiming Xie, and many other colleagues for their hard work in the field. Thank reviewers Alison M. Murray and Lionel Cavin for making very valuable suggestions for the improvement of the manuscript.

### Funding

This work is supported by the National Natural Science Foundation of China (Nos. 41862001, 41162002), the Natural Science Foundation of Guangxi (No. 2017GXNSFAA198291), and the Strategic Priority Research Program of the Chinese Academy of Sciences (No. XDA20070203). The funders had no role in study design, data collection and analysis, decision to publish, or preparation of the manuscript.

### Grant Disclosures

The following grant information was disclosed by the authors:
National Natural Science Foundation of China: 41862001, 41162002.
Natural Science Foundation of Guangxi: 2017GXNSFAA198291.
Strategic Priority Research Program of the Chinese Academy of Sciences: XDA20070203.

### Competing Interests

The authors declare there are no competing interests.

### Author Contributions

- Gengjiao Chen conceived and designed the experiments, performed the experiments, analyzed the data, prepared figures and/or tables, authored or reviewed drafts of the paper, and approved the final draft.
- Mee-mann Chang conceived and designed the experiments, performed the experiments, authored or reviewed drafts of the paper, and approved the final draft.
- Feixiang Wu conceived and designed the experiments, performed the experiments, prepared figures and/or tables, authored or reviewed drafts of the paper, and approved the final draft.
- Xiaowen Liao performed the experiments, analyzed the data, prepared figures and/or tables, and approved the final draft.

### Data Availability

The anatomical character list, data matrix, the strict consensus tree, and Tree retrieved by Bayesian phylogenetic analyses are available in the Supplementary Files.

The specimen (NHMG 005532, NHMG 033658-033661, NHMG 03368100336836, NHMG 0336859, NHMG 038777-038778, NHMG 011647-011651, NHMG 004929) illustrated and described is stored at the fossil collections of the Natural History Museum of Guangxi Zhuang Autonomous Region in Nanning, China.

## New Species Registration

The following information was supplied regarding the registration of a newly described species:

Publication LSID: urn:lsid:zoobank.org:pub:99B7F0EE-3695-4178-9606-1CD8BD90316C

Guiclupea gen. nov. LSID: urn:lsid:zoobank.org:act:DFC28EE1-F652-4961-BE7C-3DA1D4ACA9D6

Guiclupea superstes, gen. et sp. nov. LSID: urn:lsid:zoobank.org:act:CE665D25-7714-48F0-AC62-8ECB906DF465

## Supplemental Information

Supplemental information for this article can be found online at http://dx.doi.org/10.7717/peerj.11418#supplemental-information.

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
