# Peer review of "Guiclupea superstes, gen. et sp. nov., the youngest ellimmichthyiform (clupeomorph) fish to date from the Oligocene of South China"

_PeerJ, doi:10.7717/peerj.11418_

## Round 0.1 · original submission · Major Revisions

· Academic Editor

Major Revisions

I have received two reviews on your manuscript. Both of them were very positive and saw great merit in your work. I agree with them and ask you to incorporated the comments on the manuscript and resubmit a revised version.

·

Basic reporting

some grammatical suggestions made on uploaded pdf

Experimental design

fine

Validity of the findings

fine

Additional comments

Overall, well written and only minor grammatical suggestions are on the uploaded pdf.
I think the discussion needs a bit more work. Diplomystus dentatus is from Eocene Green River deposits of Wyoming, USA, and Horseshoeichthys is from Maastrictian deposits of southern Alberta, Canada. Both are from freshwater deposits. These fish would not have been able to reach the Pacific Ocean, as the uplift of the Rocky Mountains would have been well under way, and the palaeolake (Green River) and palaeostream (Horseshoeichthys) would both have been on the east side of the mountains which therefore would block them from reaching the Pacific Ocean. Current drainage of the area is the Hudson Bay or the Mississippi River, and likely palaeodrainage would have been similar.
These fish may still have been amphidromous, but perhaps leaving the lakes to spawn in connected rivers, as some fish do today. There are a number of northern Canadian lakes that contain salmonids that have two morphotypes – one that is amphidromous and the other that spends all its time in the lakes. Perhaps the diplomystines were doing something similar.

I think it is interesting that the larger ellimmichthyiforms are grouping together, but I’m not convinced by your suggesting of anadromy. Perhaps there is some other reason causing the increased size. Do you think it might instead by associated with climate differences between the Cretaceous and Eo-Oligocene?

The last paragraph of the palaeobiogeography section also needs to be reworked. You say that all ellimmichthyiform fossil localities (Fig. 1) are near to the recent coast. This is not correct. The freshwater Horseshoeichthys locality is about 600 km in a straight line from the Pacific Ocean and D. dentatus is about 1000 km from the Pacific coast, and both are on the wrong side of the Rocky Mountains. Other ellimmichthyiformes are from marine deposits, so it would be better to describe them as being found in shallow marine deposits (not ‘near the coast’). Additionally, these fossil deposits are not near the recent coast, with the Moroccan (Armigatus) and Albertan (Tycheroichthys) localities not being anywhere near the sea. This might just need a bit of rephrasing, but also a bit of rethinking in terms of the freshwater forms.

·

Basic reporting

The text is clear and well written, as far as I can tell. I corrected some spelling mistakes in the pdf and suggested some English corrections.I have suggested three references that might be helpful in the discussion. The structure of the paper is good.

Experimental design

no comment, see general comments below.

Validity of the findings

no comment, see general comments below.

Additional comments

This is an important article describing a new Oligocene ellimmichthyiform fish from southeast China, representing the youngest representative of its clade. The description is detailed and I only have a few comments / remarks noted directly on the pdf. The phylogenetic analyzes are convincing and well done.
The biogeographical discussion is useful and well done.

In addition to the comments written directly in the pdf, I have the following remarks:

- Why did you choose NHMG 005532 as your holotype? From the photograph, it appears that very little detail of the skull is visible. NHMG 033659 shows many more cranial characters and several postcranial characters.

- In phylogenetic analyzes, what is the theoretical reason for using Elops in one matrix and Chanos in another matrix? Why not use both genera as outgroups in the same matrix.

- The characters supporting the nodes are detailed for each major clade. I would suggest that the authors indicate which of these characters are uniquely derived (ci = 1) and which are known in most terminal taxa for the most important clades. This information is important because the clade can be supported by very homoplastic character states and by character states unknown in most terminal taxa, which makes these nodes very weak. I realize that this information is available in Figs 1S and 2S (and I notice that there are very few uniquely derived character states), but it would be good to mention these character states in the main text.

- Lines 527-528: "It should be noted that the Diplomystus order or clade appears to show a tendency to enlarge their body size." This trend has been observed in many clades of ray-finned fish (Guinot and Cavin, 2018) over a long time interval (but strangely enough for clupeiformes but not for ellimmichthyiformes). This is a complementary explanation to the explanation linked to migratory vs non-migratory behavior proposed by the authors. This article also discusses some general paleogeographic and paleoecological patterns of clupeomorphs.

- Regarding the last paragraph of the discussion dealing with the Pacific distribution of ellimmichthyiformes and their potential close link with the sea to explain this distribution could benefit from a comparison with the distribution of Clupeiformes, and their numerous marine / freshwater transitions, studied by Lavoué et al. (2013, 2014).

- Figures 6. Why is Gasteroclupea always included in the Sorbinichthyidae when the analysis did not resolve this group as monophyletic? Maybe this is a detail, but I wonder why the authors chose long ghost lines for the genera rather than making them as short as possible? For example, the radiation of Diplomystus and the separation with Guiclupea may have occurred during the early Upper Cretaceous, with a single ghost line reaching the basal Cretaceous, rather than 5 ghost lineages reaching the basal Cretaceous.
‘Rhombichthys’ rather than ‘Rhomichthys’ in figures 8 and 9.


References
Guinot, G., & Cavin, L. (2018). Body size evolution and habitat colonization across 100 million years (Late Jurassic–Paleocene) of the actinopterygian evolutionary history. Fish and Fisheries, 19(4), 577-597.

Lavoué, S., Miya, M., Musikasinthorn, P., Chen, W.-J., & Nishida, M. (2013). Mitogenomic evidence for an indo-west pacific origin of the clupeoidei (Teleostei: Clupeiformes). PLoS One, 8(2), e56485. https://doi.org/10.1371/journal.pone.0056485

Lavoué, S., Konstantinidis, P., & Chen, W.-J. (2014). Progress in clupeiform systematics. In K. Ganias (Ed.), Biology and ecology of sardines and anchovies (Vols. 1–0, pp. 3–42). Boca Raton, FL: CRC Press. http://www.crcnetbase.com/doi/abs/10.1201/b16682-3

Lionel Cavin

---

## Round 0.2 · accepted · Accept

· Academic Editor

Accept

I’m glad to recommend the manuscript for publication as is.

·

Basic reporting

no comment

Experimental design

no comment

Validity of the findings

no comment

Additional comments

The authors have responded to all of the comments I suggested in the first review and I consider the article to be acceptable now in its current form.
Best regards,

Lionel Cavin